# Understanding Solo Female Travellers in Canada: A Two-Factor Analysis of Hotel Satisfaction and Dissatisfaction Using TripAdvisor Reviews

Feiyan Zhou [ID], Shuyue Huang *[ID] and Maria Matthews

Business and Tourism Department, Mount Saint Vincent University, Halifax, NS B3M 2J6, Canada; yanfa0176@gmail.com (F.Z.); maria.matthews2@msvu.ca (M.M.)
* Correspondence: shuyue.huang@msvu.ca

**Abstract:** This study aims to understand solo female travellers' hotel experiences in Canada by analyzing online reviews from TripAdvisor. We employed keywords such as "solo female" and "single female" to identify online reviews, followed by a manual review process to confirm their relevance and eliminate duplicates. The final dataset included 240 reviews from 188 lodging establishments, totalling 49,924 words. Employing Herzberg's two-factor theory and NVivo, we generated codes and categorized them into 29 satisfiers and 24 dissatisfiers. These were grouped into five key components impacting guests' experiences: room, staff, hotel facilities and cleanliness, hotel amenities, and others. The top three satisfiers identified in traditional accommodations are safety, staff helpfulness, and location, while room dirtiness, insecurity, and room amenities are the primary dissatisfiers. Conversely, alternative lodgings reveal a distinct pattern, with location, room amenities, and staff friendliness as top satisfiers, and room amenities, neighbourhood, and service unavailability as leading dissatisfiers. The study found that alternative accommodations may offer a broader range of experiences, potentially due to their less-standardized nature and diversity of options. This research enhances understanding of solo female travellers, gender differences in hotel experiences, and customer satisfaction, underscoring the tourism industry's need to address this demographic's unique needs and concerns.

**Keywords:** sole female travellers; TripAdvisor reviews; satisfiers; dissatisfiers; hotel experience; Canada

## 1. Introduction

Solo travel has emerged as a significant trend [1,2], with studies increasingly focusing on both solo female [3,4] and male travellers [5] in recent years. Specifically, women have transitioned from their traditional roles as decision makers within family and group travel contexts [6] to orchestrating their own travel endeavors with increasing frequency [7]. The phenomenon of females travelling alone has drawn attention from both the tourism industry and academia. Industry pioneers have witnessed a significant increase in demand; as of May 2022, The Solo Female Traveler Network has recovered its business to 60% of what the pre-COVID level was, and G Adventures has over half of their bookings from solo travellers, and among which 70% are women [8]. The solo travel segment is rapidly expanding, evidenced by projections indicating it as the fastest-growing segment with an expected CAGR of 7.5% from 2023 to 2032 [9]. Additionally, a marked increase in solo travel demand is supported by data from Booking.com, which shows a rise from 14% pre-pandemic to 23% of travellers going solo by mid-2021, and a significant 761.15% increase in solo travel searches according to Google trend data [10]. Researchers from around the world have studied solo female travellers from different source markets, such as China [11], Vietnam [12], and Iran [4] and travelling to different destinations, like India [13] and Australia [14]. The current studies on solo female travel have also explored the issues of motivations and constraints [12,15,16] and safety and risk concerns [17].

Yang [16] offers valuable insights into solo travel, exploring definitions, motivations, and constraints, and particularly emphasizes the underexplored domain of solo female travellers, despite growing interest in the area. More specifically, Yang's work highlights that current business operators predominantly cater to couples, families, or groups, often overlooking the unique experiences of solo travellers, especially in the accommodation sector. Concurrently, Otegui-Carles et al. [2], in their bibliometric review of solo travel research, advocate for broadening the scope of analyses beyond traditional aspects such as motivations, experiences, and constraints, and for expanding research to regions beyond Asia. They also highlight the importance of focusing on solo female travellers due to their rapidly growing numbers within the tourism sector and the need to promote gender equality in tourism. Building upon Yang's [16] insights into the overlooked domain of solo travellers' experiences in the accommodation sector and Otegui-Carles et al.'s [2] call for a broader research scope, the existing literature could benefit from exploring concepts such as customer satisfaction and dissatisfaction beyond traditional aspects, with a focus on solo female travellers, particularly regarding experiences outside of Asia. Our research seeks to bridge these gaps by shedding light on the Canadian market, specifically targeting the burgeoning demographic of solo female travellers within the accommodation sector. In doing so, we aim to identify the determinants of their satisfaction and dissatisfaction, contributing to a more inclusive understanding of solo travel and advancing efforts toward gender equality in tourism.

It is worth noting that gender differences impact not only how hotel service quality predicts guests' satisfaction and loyalty [18] but also the way guests articulate instances of service failures in their reviews [19]. Using evidence from TripAdvisor, Rahimi et al.'s [20] study demonstrated critical differences between female and male guests' preferences towards hotel attributes and services: women pay greater attention to the hotel's core offerings and their comfort when evaluating hotel service, while when it comes to food and beverages, men show greater interests in discussing pubs, beer, and particular types of cuisine. Recognizing the importance of understanding solo travellers' accommodation experiences and the distinct needs of different genders, our study concentrates on the factors influencing solo female travellers' satisfaction and dissatisfaction with various types of lodgings and is split into two categories of traditional and alternative lodging.

Customer satisfaction is essential for the success of hospitality organizations. It fosters positive emotions such as brand love [21] and encourages recommendations and intentions to revisit [22,23]. Delivering guest services in lodging establishments involves a meld of tangible and intangible elements [24]. These constituents contribute to both positive and negative outcomes, potentially leading to guest satisfaction or dissatisfaction [25]. Intriguingly, empirical studies indicate that the ramifications of customer dissatisfaction in service businesses can outweigh the positive impacts of satisfaction [26]. Dissatisfaction can beget negative behavioural intentions, such as service-provider switching, spreading negative word-of-mouth comments, and lodging complaints [27,28].

Scholars have probed customer satisfaction in hotel experiences through a variety of theoretical lenses and frameworks, including using expectancy–disconfirmation theory [29], customer delight [30], two-factor theory [31], and three-factor theory [32]. For the purpose of this study, we specifically employ the two-factor theory [33], an analytical framework that empirical research has affirmed is effective in distinguishing between the factors driving satisfaction and dissatisfaction. The two-factor theory posits that different dimensions independently influence satisfaction and dissatisfaction [34]; it distinctly addresses both positive and negative aspects of the customer experience, allowing for a comprehensive analysis of satisfiers and dissatisfiers [31]. Such a framework is particularly beneficial for hotels to address dissatisfied customer experiences effectively and remain competitive in the niche market of solo female travel, considering the unique challenges faced by solo female travellers, including safety concerns [16], sensitivity to service quality [19], and the desire for empowerment and autonomy [1,12]. We argue that such a theory can further

enlighten our understanding of the unique experiences of solo female travellers in both traditional and alternative lodging contexts.

We aim to identify the satisfiers and dissatisfiers of solo female travellers' accommodation experiences using online reviews posted on TripAdvisor. Substantial evidence has supported the use of online reviews in tourism and hospitality research [20,35]. To the authors' best knowledge, no studies have investigated solo female travellers' accommodation experience in Canada. The study's results are expected to contribute to the literature on solo female travellers, gender differences in hotel experience, and customer satisfaction. It is worth noting that the authors do not imply that the solo male experience is insignificant; rather, this study highlights the necessity for further research to comprehend the unique aspects of various subgroups, including both male and female solo travellers within the lodging sector.

## 2. Literature

### 2.1. Solo Female Travellers

The increasing prevalence of living alone has become a new norm and is accompanied by the rise of solo consumption [36], which includes activities such as travelling solo. Laesser et al. [37] suggested segmenting solo travel into four types based on a combination of (1) departure status (either a person from a single-person household or a multi-person household) and (2) arrival status (travelling solo or in a group), thus generating a two-by-two matrix, comprising four distinct segments in total. Most studies define solo travellers according to their arrival status [1], and there has been a greater focus on female solo travellers in the existing research on this topic. Solo travellers are individuals, regardless of their marital status, who opt to travel alone during their vacations [1]. According to Yang [16], solo travel can be divided into two groups, solo "by circumstances'" or "by choice". The former refers to people who do not have travel partners [38], while the latter emphasizes the individuals' choice to travel alone despite having a partner [12].

Market strategies can only be effective when customers are segmented into heterogeneous groups which respond to market stimuli differently. Thus, researchers also study the differences between solo vs. non-solo travellers [16], and solo female vs. male travellers [11]. For example, when comparing the motivations and service-quality expectations of solo female travellers and general travellers in the context of museums, Duantrakoonsil et al. [39] found that the former group was primarily driven by escapism, learning, and curiosity, whereas the latter was more motivated by social and family interactions. In addition, Radojevic et al. [40] differentiated solo travellers from other demographic groups, highlighting their preference for smaller, affordable accommodations in convenient locations with free Wi-Fi, while in the meantime, hotel classifications and brands are less important for them compared to couples and families. Concerning service quality, staff services and exhibition experiences had a more positive relationship with solo female travellers, while general travellers considered facilities to be the most crucial element [39].

Studies on solo female travellers primarily explore their motivations and constraints [1]. Within the literature, there is a specific focus on the challenges and benefits that they experience [17] and also factors determining their travel intentions [41]. For instance, Wilson and Harris [42] stated that solo travel allows women to reassess their viewpoints on life and society, encompassing their relationships with others as well. Osman et al.'s [12] study on Vietnamese solo female travellers concluded with two categories of motivations: personal factors, including freedom and flexibility, self-empowerment, independence, and exploration; social interaction factors, such as lack of companions and the opportunity to meet new people. More specifically, they are inspired by the opportunity to seek freedom, self-development and self-enhancement, being autonomous through travelling solo, and connectedness with people [1,12,43]. Similarly, Breda et al.'s [44] study, which focused on Portuguese women, identified the key motivations of solo female travellers as including the absence of a travel companion, freedom of choice, the pursuit of adventure and experience, and escaping from the daily routine. Their study also indicated that most women who

travel alone are young, single, and childless, predominantly identifying as adventurous, outgoing, and independent. However, the profiles of solo female travellers may vary depending on the participants interviewed by diverse researchers. Solo travel also allows females to have a more profound cultural exchange with local communities [14]. It can also positively impact individuals' well-being, as solo travel helps triumph over stress and depression, escape daily responsibilities, and brings happiness, empowerment, self-growth, and self-realization [3].

Yang [16] suggested three barriers to solo travel participation and experience: safety, cost, and social constraints. For instance, risk and safety issues are related to the stigma of women staying at home instead of risking their lives to travel alone [12], negative encounters with male strangers [45], and the biological differences between the sexes, which result in varying perceptions of vulnerability and risk-taking behaviours during solo travel. Additionally, Breda et al. [44] identified the main challenges faced by women when travelling solo as loneliness, harassment, and the fear of walking alone at night or being robbed. Economic constraints can also arise when travelling alone. For instance, the single supplement, a fee charged by most hotels for solo travellers, is perceived as a primary constraint [1]. Solo travel might require extra accommodation expenses and result in more complaints if the lodging industry business operators only focus on couples, families, or groups [16]. Solo travellers may also experience dissatisfaction due to poor infrastructure, the need to address issues independently, and coping with illness while travelling alone [1]. Regarding social constraints, Asian women's travel experiences have some unique circumstances related to the Asian socio-cultural context, despite the increasing number of people travelling alone [7].

Beyond the general challenges of solo travel, one particular issue that needs special attention is the sexualization solo female travellers often confront [45]. For instance, Jordan and Aitchison [46] discuss how solo female travellers can be subjected to the "sexualization of the gaze", with local men sometimes misinterpreting their solo presence as an open invitation, leading to instances of unwelcome attention and harassment. In destinations known for sex tourism, Asian female backpackers encounter a "double jeopardy" due to racial and gender stereotypes, challenging the normative image of the independent traveller as white and male [47]. Yang et al. [17] highlight that Asian solo female travellers face both gendered (such as sexual assault and street harassment) and racialized (including discrimination and social disapproval) risks. Their study illustrates how these travellers perceive and navigate these risks, leading to empowerment and self-transformation. These experiences underscore the compounded safety concerns for solo female tourists, highlighting the critical need for both academia and the tourism industry to implement gender-sensitive approaches that respect and protect solo female tourists, thereby advancing gender equity in travel.

### 2.2. Gender Differences in Hotel Experiences

Empirical studies have demonstrated the importance of considering customers' expectations and perspectives from a gender viewpoint in order to enhance the service quality in the hotel industry [48,49]. Building on this notion, previous research has delved into uncovering gender disparities specifically within the context of hotel experiences [19,20,50]. For instance, Bogicevic et al.'s [51] study revealed distinct preferences between female and male guests regarding hotel room design, showing that females were equally content with rooms featuring either masculine or feminine colour schemes, whereas males favoured rooms decorated with masculine colours.

Kim et al. [52] examined the effects of temporal distance and gender differences on how individuals perceive the importance of various factors when choosing a hotel. Their findings revealed a significant main effect of gender, with women assigning more importance to factors such as employee service, value for money, and room quality; additionally, they identified a significant interaction effect between temporal distance and gender for the hotel-selection process. Additionally, drawing from online customer reviews of urban

hotels, Sánchez-Franco et al. [50] discovered gender differences in the perception of the importance of hotel services; specifically, males favour strategies based on easily accessible cues and prioritize room features (e.g., appearance, air conditioning, noise levels, cleanliness), while females may have significantly lower expectations regarding these room features.

It is worth noting that additional studies are needed to amplify women's voices in various diverse accommodation settings. For instance, Farmaki [53] highlights the importance of understanding women's motivations for engaging in peer-to-peer (P2P) accommodation, emphasizing that both solo female travellers and female hosts are not immune to gendered risks. Issues of safety, trust, and privacy emerge as critical concerns for both parties [53]. Therefore, it is imperative to examine female travellers' experiences across diverse accommodation settings, which is also the focus of the current study.

### 2.3. Satisfiers and Dissatisfiers

Customer satisfaction means "the consumer's fulfillment response. It is a judgment that a product or service feature, or the product or service itself, provided (or is providing) a pleasurable level of consumption related fulfillment, including levels of under- or over-fulfillment" [54] (p. 13). It is an essential factor to measure a company's performance [55] and is directly associated with post-consumption behaviours such as repurchase intention, word of mouth [22], and brand attitudes [56].

Herzberg's two-factor theory, or Motivator and Hygiene Factor Theory [33] was originally developed to identify mutually exclusive factors (i.e., motivator/satisfiers and hygiene/dissatisfiers) that determine job satisfaction. This theory posits that the dimensions of satisfaction and dissatisfaction are connected to job satisfaction through two categories of factors: hygiene factors (dissatisfiers), which result in job dissatisfaction, and motivators (satisfiers), which contribute to achieving a state of satisfaction [33]. In consumer behaviour studies, dissatisfaction results from dissatisfiers or the lack of hygiene factors; however, it does not strengthen satisfaction and, the absence of satisfiers does not necessarily cause dissatisfaction [57,58].

Specifically, the two-factor theory has been adopted in understanding customer satisfaction in tourism and hospitality management (e.g., [31,59]). Balmer and Baum [60] found that satisfiers are primarily related to intangible components (e.g., recognition by staff, sense of belonging, flexibility by the hotel, and service orientation), whereas dissatisfiers are more connected to tangible aspects (e.g., pricing, facilities–cleanliness, size, variety, and freebies/extras). Kim et al. [31] identified different satisfiers and dissatisfiers in full-service and limited-service hotels in New York City and suggested hotel class should be considered when implementing strategies to achieve customer satisfaction; also, the "staff and their attitude" was identified as the most important factor. Luo and Qu [61] identified 10 attributes of hotel services that significantly influence guest satisfaction or dissatisfaction. Specifically, service delivery, environment, and price were recognized as satisfiers, contributing to guests' overall satisfaction; on the other hand, cleanliness, internet connectivity, employee attitudes, facilities, security, location, and food and beverages were classified as dissatisfiers, suggesting that while their impact on guest satisfaction may be limited, they can cause dissatisfaction when they are poorly delivered [61].

Examples of utilizing Herzberg's two-factor theory can also be observed in other hospitality sectors. For instance, Chan and Baum [34] applied Herzberg's two-factor theory in the context of ecolodge service consumption, distinguishing satisfiers as being derived from intangible elements, while dissatisfiers originate from tangible elements. Using customer reviews from TripAdvisor, Park et al. [59] distinguished satisfiers and dissatisfiers of the quality of airline service attributes: the former contains cleanliness, food and beverages, and in-flight entertainment, and the latter includes customer service and check-in and boarding. In other words, food and beverages, as well as in-flight entertainment, are crucial factors in determining positive ratings (satisfaction) but have minimal impact on explaining variations in negative ratings (dissatisfaction) [59]. Consequently, conclusions regarding

the identification of satisfiers and dissatisfiers may not be universally applicable due to the varying types of service attributes present in different industry sectors. It is crucial to differentiate between satisfiers and dissatisfiers for solo female travellers in lodging experiences, as these factors may vary based on their chosen accommodations and the different levels of customer expectations associated with each type of accommodation.

While we adopted the two-factor theory in this study for its unique ability to discern between hygiene factors and motivators, it is important to recognize that various satisfaction theories have their merits and have been proven effective in empirical studies (e.g., [29,30,32]). Our focus on this theory does not diminish the potential applicability of other frameworks in exploring different dimensions of satisfaction and suggests the opportunities for applying them to enrich the solo female travellers' hotel experiences. For instance, expectancy–disconfirmation theory focuses on the gap between expected and actual service experiences, highlighting the risk of setting high expectations, especially where higher prices might lead to greater disappointment if they are not met [29]. This theory is instrumental in understanding the dynamics between expectations, actual performance, and the subsequent confirmation or disconfirmation of those expectations, offering insights into their collective impact on customer satisfaction [62]. Furthermore, customer delight, which transcends mere satisfaction by exceeding expectations, involves elements of joy and surprise [63] and is particularly relevant in emotionally engaging settings like upscale hotels [64] and theme parks [65]. Additionally, the three-factor theory [66], adds hybrids as a third category of service attributes, providing a deeper understanding of how various service elements affect customer satisfaction [67].

## 3. Materials and Methods

### 3.1. Data Collection

Studies have provided substantial evidence supporting TripAdvisor as an influential platform for authentic customer reviews [20,31,35]. This study examined reviews posted on TripAdvisor under the "Hotels" section within the context of Canada. Canada ranked 13th in the 2021 Travel & Tourism Development Index, securing a position among the top global rankings out of 117 economies [68]. Moreover, a recent study indicated that Canada ranked 8th globally for solo female travel among 34 countries analyzed [69]. These accomplishments make Canada a competitive destination for international and domestic tourism, catering to solo female travellers as well.

By conducting a search for keywords like "solo female" and "single female" in TripAdvisor reviews of Canadian hotels, the initial step yielded a total of 364 downloaded reviews from Canadian hotels as of 30 September 2022. A manual review step was further employed to ensure all reviews were relevant to the research topic and to eliminate duplicate reviews. The step resulted in a total of 240 reviews from 188 lodging establishments, containing a total of 49,924 words. On average, each review consisted of 208 words. Please note that each data record collected included details on the hotel name, website, address, province, hotel class, ID of the reviewers, review title and content, and a rating ranging from 1 to 5. The reviews left by guests on the TripAdvisor website aim to provide an overall evaluation of their hotel stay using a five-point Likert scale, ranging from "terrible" (1) to "poor" (2), "average" (3), "very good" (4), and "excellent" (5). This rating is a trustworthy assessment for this study as guests can provide feedback on one of the five-point ratings, reflecting their overall hotel experience.

Before conducting the data analysis, the lodging accommodations were classified into two categories: traditional hotels and alternative lodgings based on a variety of factors, such as service level, service type, facility quality, ambiance, and price. It is worth noting that the distinction between these two categories is not widely accepted or clearly defined in the academic literature. However, most studies have focused on traditional hotel settings, such as using the five-star rating system or full-service versus limited-service hotels. Although TripAdvisor offers hotel-class star ratings from national rating agencies and third-party partners like Giata, many alternative lodgings do not have a clear hotel class [70]. For the

purpose of this study, hotel chains, international brand hotels, and traditional hotels were classified as traditional accommodations, while B&Bs, university residences, camping sites, hostels, glamping units, and other accommodations were classified as alternative lodgings.

### 3.2. Data Analysis

The reviews were categorized as either reflecting satisfaction or dissatisfaction. Satisfiers were identified from positive reviews marked as "excellent" and "very good", referred to as satisfaction-indicating reviews. Conversely, dissatisfiers were discerned from negative comments labelled as "terrible" and "poor", referred to as dissatisfaction-indicating reviews. Reviews with an average rating were not included in the analysis, as they contained mixed information and did not clearly indicate a positive or negative experience, aligning with the specific focus of this study.

In terms of the data analysis, we employed a content analysis method [71], using NVivo to generate codes, which were further categorized into satisfiers and dissatisfiers according to Herzberg's two-factor theory [33]. This included a software-assisted text-mining analysis and a manual holistic approach, supported by recent studies in hospitality research [31,72] that utilize online reviews to identify significant factors. However, since satisfaction-indicating reviews with ratings of 4 and 5 may include negative aspects and vice versa, a complete computerized analytical program might overlook these subtleties. Consequently, a thorough manual review process was employed to manually code all the reviews with reference to the literature on hotel services and attributes. The process involved one author coding and the other two authors validating, and the codes were subsequently categorized into two groups, namely satisfiers and dissatisfiers. In our data analysis, we utilized a bottom-up, inductive approach recommended by Thomas [73], which facilitated the emergent coding of data into satisfiers and dissatisfiers. Subsequently, these initial codings were aggregated into broader themes, allowing for a nuanced interpretation of the data. This method aligns with the practices of qualitative data analysis in the existing literature [74,75], offering the advantage of uncovering nuanced textual features that might have been previously overlooked. It is important to note that our thematic labelling was influenced by the literature review conducted prior to data collection and analysis, ensuring that our themes were both grounded in and contribute to the existing knowledge on key hotel attributes and services that influence customer satisfaction and dissatisfaction.

Last but not least, the satisfiers and dissatisfiers within the key themes were also compared between alternative and traditional hotels. This distinction emphasizes the nuanced differences in customer preferences and aligns with the broader objectives of the study, providing insight into varying accommodation experiences across different hotel types.

## 4. Results

### 4.1. Classification of Online Reviews

After compiling all of the online reviews, the data revealed that there were 130 reviews from 87 alternative lodgings and 110 reviews from 101 traditional accommodations. The reviews were categorized according to their ratings, with satisfaction-indicating reviews derived from ratings of 4 and 5, and dissatisfaction-indicating reviews from ratings of 1 and 2. Reviews with a rating of 3 were excluded from the analysis. We acknowledged the possibility of divergent sentiments within a single review; that is, reviewers who categorized their overall experience as 'excellent' may have also included negative remarks and vice versa. However, for the integrity of our data analysis, we meticulously examined each review to isolate and consider only the comments that aligned with the overall sentiment. Consequently, we centered our attention solely on identifying satisfiers within the positive review categories and, likewise, dissatisfiers from the negative categories.

Table 1 provides a detailed breakdown of the review data for both alternative and traditional lodgings. For alternative lodgings, the analysis included 97 satisfaction-indicating reviews, encompassing 20,072 words (an average of 207 words per review), and 20 dissatisfaction-

indicating reviews, totalling 5199 words (an average of 260 words per review). This made a sum of 117 reviews specifically for alternative lodging. A notable disparity is evident in the number of reviews used to isolate dissatisfiers compared to satisfiers, suggesting a greater propensity for satisfaction in alternative lodgings.

**Table 1.** Summary of review data for alternative and traditional lodgings.

| Lodging Category | Review Rating | No. of Reviews | Review Classification | Total |
|---|---|---|---|---|
| Alternative lodgings (n = 87) | 1 | 9 | Dissatisfaction-indicating | 130 |
| | 2 | 11 | | |
| | 3 | 13 | Excluded | |
| | 4 | 37 | Satisfaction-indicating | |
| | 5 | 60 | | |
| Traditional accommodations (n = 101) | 1 | 28 | Dissatisfaction-indicating | 110 |
| | 2 | 13 | | |
| | 3 | 22 | Excluded | |
| | 4 | 18 | Satisfaction-indicating | |
| | 5 | 29 | | |

Note: In total, 61 dissatisfaction-indicating reviews, and 144 satisfaction-indicating reviews.

In the case of traditional hotels, the dataset for analyzing satisfiers and dissatisfiers was equally balanced. It consisted of 47 satisfaction-indicating reviews, with a total of 7021 words (an average of 149 words per review), and 41 dissatisfaction-indicating reviews, amounting to 8770 words (an average of 214 words per review). This resulted in a total of 88 reviews utilized for the data analysis of traditional accommodation.

In alternative lodgings (Figure 1), 79.6% of the 97 satisfaction-indicating reviews were entirely positive, while 20.4% were mixed. Of the 20 dissatisfaction-indicating reviews, 55% were solely negative, with the remaining 45% containing some positive input within predominantly negative feedback. In traditional accommodations, among the 47 satisfaction-indicating reviews, 80.6% were purely positive and 19.4% mixed. Additionally, we analyzed 41 dissatisfaction-indicating reviews, with 75.5% being entirely negative and 24.5% mixed, to identify key dissatisfiers.

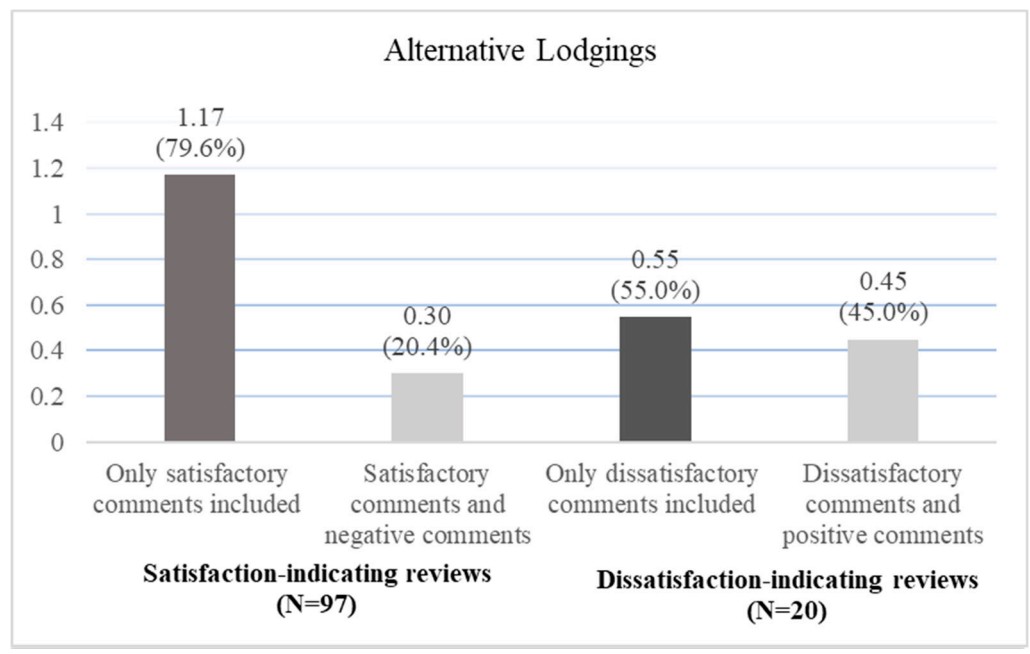

**Figure 1.** *Cont*.

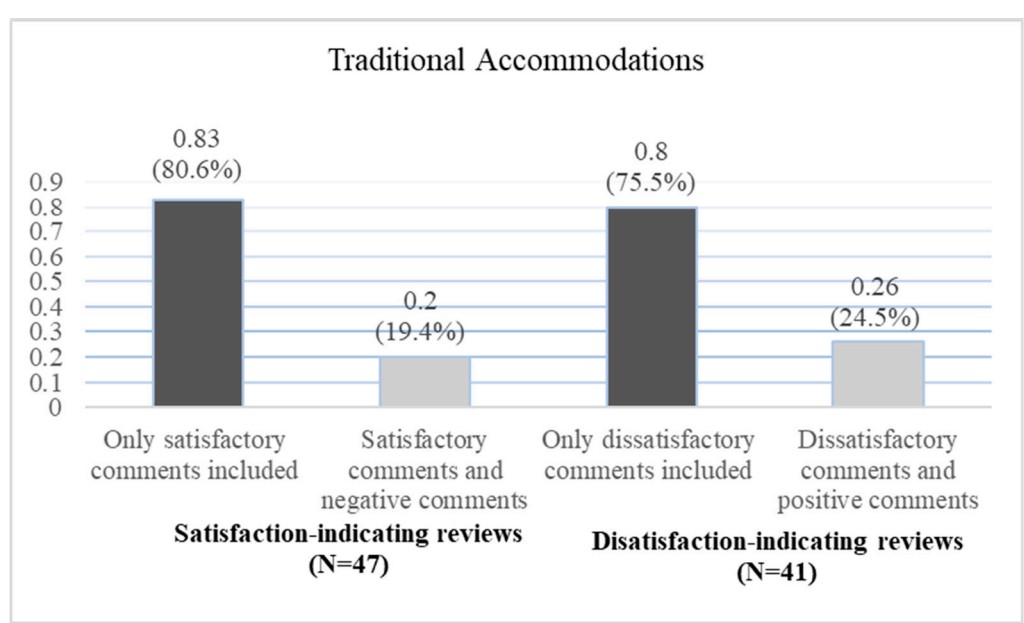

**Figure 1.** Distribution and average count of satisfaction-indicating and dissatisfaction-indicating reviews.

Displayed above each column in Figure 1 are numbers denoting, on average, the number of reviews per hotel within specific categories. For instance, each hotel receives approximately 1.17 purely positive satisfaction-indicating reviews, as opposed to an average of 0.30 mixed reviews per hotel—reviews that contain both positive and negative sentiments in alternative lodgings.

*4.2. Identifying Satisfiers and Dissatisfiers in Traditional and Alternative Lodgings*

The satisfiers and dissatisfiers identified in this research were organized into five crucial components shaping the guest's hotel experience. These components encompassed room, staff, hotel facilities and cleanliness, hotel amenities, and others, as shown in Table 2. This structure was derived from a modified categorization of hotel attributes previously employed in scholarly studies [19,31,50,52,61], recognizing that such categorizations can fluctuate depending on various factors, such as the type of hotel under investigation [31], the guest's gender [19], and the hotel design [51].

**Table 2.** Five key components of satisfiers and dissatisfiers in lodging reviews.

| Room | Staff | Hotel Facilities and Cleanliness | Hotel Amenities | Others |
|---|---|---|---|---|
| **1.1 Room size** **1.2 Room amenities** **1.3 Room cleanliness (Dirtiness)** **1.4 Bed comfort** **1.5 Room quietness (Noisiness)** **1.6 Room design and condition** **1.7 Room assignment** | **2.1 Staff friendliness (Unfriendliness)** **2.2 Staff helpfulness (Unhelpfulness)** **2.3 Staff professionalism (Unprofessionalism and harassment)** *2.4 Service unavailability* | **3.1 Hotel facilities** **3.2 Hotel cleanliness (Dirtiness)** **3.3 Hotel property appearance** | **4.1 Wi-Fi** **4.2 Breakfast** **4.3 Parking** **4.4 Food** 4.5 Lockers 4.6 Recreation and entertainment activities 4.7 Other amenities | 5.1 Location 5.2 Transportation 5.3 View **5.4 (Poor) Management and policy** **5.5 Value for money** **5.6 Safety (Insecurity)** **5.7 Neighbourhood** **5.8 Roommates and fellow guests** 5.9 Host |

Notes: A combined analysis of the reviews from both alternative and traditional lodgings was conducted, grouping various satisfiers and dissatisfiers into five key components. Words of parentheses belong to interchangeable language. The items that are bold are represented in both the satisfier and dissatisfier categories. Underlined items exclusively belong to the satisfiers category, while items in *italics* are unique to the dissatisfiers category.

The room component was constituted by seven distinct attributes: room size, room amenities, room cleanliness (and its negative counterpart, dirtiness), room quietness (or noisiness), room design and condition, and room assignment. Interestingly, these seven attributes surfaced in both the satisfier and dissatisfier categories, illustrating their dual role in shaping guests' hotel experiences. These attributes have been proven crucial in the literature due to the substantial amount of time guests spend in their rooms [51], which directly impacts their overall experience and sleep quality [76]. Our identification of the room component aligns with the existing research [77], signifying its vital role in shaping the hotel experience. For instance, previous studies have identified quiet rooms [19] and comfortable beds [31] as significant contributors to guest satisfaction. On the flip side, the absence of these elements, such as noisy environments or uncomfortable beds, could transform the room component into a dissatisfier.

The staff component encapsulates four key attributes: staff friendliness (or its negative opposite, unfriendliness), staff helpfulness (and its dissatisfactory counterpart, unhelpfulness), staff professionalism (or its opposing state, unprofessionalism and harassment), and the absence of service (service unavailability). It is important to differentiate between service unavailability, which denotes a complete absence of service, and staff unhelpfulness, where the service exists but is not efficiently provided or helpful. The first three attributes emerge within both the satisfier and dissatisfier groups, with the addition of service unavailability in the latter. Hotel staff significantly shape guest experiences, playing a critical role in delivering services and establishing the quality of guest–staff interactions [32]. For instance, when staff members exude friendliness and demonstrate a willingness to solve problems, guests often report higher levels of satisfaction. This emphasis on staff resonates with findings from Kim et al. [31], who concluded that, in both full-service and limited-service hotels, staff and their attitude stand as the most critical factor in satisfying customers and addressing dissatisfaction. Such observations align with another study by Huang et al. [19], which found that female guests, in particular, place significant value on attitudes and affections. Disrespect or lack of attention is more likely to create a negative impression among female guests, as they tend to pay more heed to interpersonal relationships and have higher expectations of staff communication skills [49].

The hotel facilities and cleanliness component encompasses three key attributes: hotel facilities, cleanliness, and property appearance. Facilities, including fitness centers, spas, and dining options, significantly impact guest satisfaction, particularly for solo female travellers seeking amenities for comfort and safety. Cleanliness, which encompasses both sanitation and odors, is a critical aspect demanding high standards in maintaining a well-kept establishment. Females, in particular, may be more sensitive to smell [19]. However, cleanliness is an essential need for all hotel guests, regardless of gender or purpose of stay [19]. Failure to meet these standards can lead to dissatisfaction, as confirmed by previous studies [60,61]. The hotel's property appearance, comprising architecture, decor, and maintenance, shapes a guest's impressions and overall perception of the quality. Overall, this component aligns with service quality dimensions in previous studies [78,79], which emphasize facilities and environments as essential factors for service quality to determine customer satisfaction. Moreover, it resonates with Luo and Qu's [61] study, which identified environment (satisfier), facilities (dissatisfier), and cleanliness (dissatisfier) as three of the ten critical factors of service quality. Our approach combines these attributes, recognizing them as both satisfiers and dissatisfiers, to provide a comprehensive understanding of their impact on solo female guest experience.

The hotel amenities component includes seven items: wifi, breakfast, parking, food, lockers, recreation and entertainment activities, and other amenities. Wifi is essential for connectivity, especially for solo female travellers, for whom internet access may be the primary means of gathering information. This aligns with Radojevic et al.'s [40] findings, as for solo travellers, they rely more on electronic communication and entertainment due to the absence of companionship. While wifi availability is generally assumed, most comments in online reviews focus not on its availability but on the quality of the wifi service, including

aspects like signal strength and reliability. These aspects significantly influence guest satisfaction and their overall experience. A well-executed breakfast and convenient parking add value, but poor implementation can lead to dissatisfaction. Food quality, options, and dietary accommodations are vital, and failures here may result in dissatisfaction. Lockers (in alternative lodgings) provide security, recreation and entertainment activities enhance value, and other amenities can improve the overall experience. Our study identified all seven items as satisfiers, with four—wifi, breakfast, parking, and food—also acting as dissatisfiers if they are poorly managed. This emphasizes the need for quality delivery to meet guests' expectations, particularly for solo female travellers.

The final component, 'others', comprises nine distinct aspects: location, transportation, view, (poor) management and policy, value for money, safety (or insecurity), neighbourhood, roommates and fellow guests, and host. However, in the dissatisfier category, the host element is excluded. Location influences satisfaction based on a hotel's proximity to attractions and transportation. Transportation accessibility enhances satisfaction, while difficulties in this area can lead to dissatisfaction. View refers to the visual appeal from the lodging, with scenic views enhancing experiences and poor ones causing dissatisfaction. Management and policy influence guest perceptions, and value for money becomes particularly vital for solo female guests, who have no one to share accommodation costs with and prefer more affordable options [40]. Safety is crucial for solo female travellers, influencing their comfort and willingness to explore. The neighbourhood's surroundings and accessibility determine whether guests feel secure enough to venture out at night or explore confidently. Unlike family or group travellers, who primarily interact with their companions, solo female travellers often engage with fellow guests, roommates, or the host in shared accommodations. This interaction aligns with existing studies on motivations, such as social interaction factors [12], the opportunity for connectedness with people [43], or the challenge of loneliness [44]. It is also a cost-saving or risk-mitigating strategy, emphasizing the importance of the social aspect of their lodging experience. The components identified align with Yang [16], who identified three barriers to solo travel participation and experience: safety, cost, and social constraints.

To summarize, our review analysis identified specific satisfiers and dissatisfiers across five key components of guest experience. We uncovered a total of 906 instances of satisfiers and 241 instances of dissatisfiers throughout all the analyzed reviews. From 144 satisfaction-indicating reviews, we found 29 satisfiers: 7 in the room component, 3 in staff, 3 in hotel facilities and cleanliness, 7 in hotel amenities, and 9 in others. From the 29 unique satisfiers, 22 appeared in reviews of traditional accommodations (with 270 mentions), and all 29 emerged in reviews of alternative accommodations (accounting for 636 mentions).

Similarly, from 61 dissatisfaction-indicating reviews, we discerned 24 dissatisfiers: 9 in room, 4 in staff, 3 in hotel facilities and cleanliness, 4 in hotel amenities, and 6 in others. These findings present critical insights for enhancing the hotel experience of solo female travellers. In terms of the 24 unique dissatisfiers, all were found in traditional accommodation reviews (154 mentions), whereas only 20 were mentioned in alternative accommodation reviews (totaling 87 mentions).

### 4.3. Satisfiers and Dissatisfiers in Traditional Accommodations

We further identified the top 10 satisfiers and dissatisfiers separately for both traditional and alternative accommodations. These findings are presented in Table 3 (Top 10 satisfiers and dissatisfiers in traditional accommodations) and Table 4 (Top 10 satisfiers and dissatisfiers in alternative lodgings). This detailed categorization offers granular insights into guest experiences across varied accommodation types.

**Table 3.** Top 10 satisfiers and dissatisfiers in traditional accommodations.

| | | | | | Traditional Hotel (N = 101) | | | | |
|---|---|---|---|---|---|---|---|---|---|
| Rank | Satisfiers | Hotel Components | Frequency | Percentage | Rank | Dissatisfiers | Hotel Components | Frequency | Percentage |
| 1 | Safety | O | 25 | 9.3% | 1 | Room dirtiness | R | 26 | 16.9% |
| 2 | Staff helpfulness | S | 24 | 8.9% | 2 | Insecurity | O | 18 | 11.7% |
| 2 | Location | O | 24 | 8.9% | 3 | Room amenities | R | 16 | 10.4% |
| 3 | Room cleanliness | R | 20 | 7.4% | 4 | Staff unprofessionalism and harassment | S | 13 | 8.4% |
| 3 | Staff friendliness | S | 20 | 7.4% | 5 | Room condition | R | 11 | 7.1% |
| 4 | Breakfast | HA | 18 | 6.7% | 5 | Neighbourhood | O | 11 | 7.1% |
| 5 | Room amenities | R | 17 | 6.3% | 6 | Poor management and policy | O | 9 | 5.8% |
| 6 | Bed comfort | R | 16 | 5.9% | 7 | Service unavailability | S | 8 | 5.2% |
| 7 | Value for money | O | 14 | 5.2% | 7 | Hotel property and appearance | HFC | 6 | 5.2% |
| 8 | Hotel facilities | HFC | 13 | 4.8% | 8 | Staff unfriendliness | S | 6 | 3.9% |
| Total | 22 | R:3, S:2, HFC:1, HA:1, O:3 | 191 | 70.8% | Total | 21 | R:3, S:3, HFC:1, O:3 | 124 | 81.7% |

Note: Five hotel components (R: room, S: staff, HFC: hotel facilities and cleanliness, HA: hotel amenities, O: others).

**Table 4.** Top 10 satisfiers and dissatisfiers in alternative lodgings.

| | | | | | Alternative Hotel (N = 87) | | | | |
|---|---|---|---|---|---|---|---|---|---|
| Rank | Satisfiers | Hotel Components | Frequency | Percentage | Rank | Dissatisfiers | Hotel Components | Frequency | Percentage |
| 1 | Location | O | 54 | 8.5% | 1 | Room amenities | R | 12 | 13.8% |
| 2 | Room amenities | R | 49 | 7.7% | 2 | Neighbourhood | O | 10 | 11.5% |
| 3 | Staff friendliness | S | 47 | 7.4% | 3 | Service unavailability | S | 9 | 10.3% |
| 4 | Hotel facilities | HFC | 39 | 6.1% | 4 | Roommates and fellow guests | O | 7 | 8.0% |
| 5 | Safety | O | 38 | 6.0% | 4 | Staff unprofessionalism and harassment | S | 7 | 8.0% |
| 6 | Breakfast | HA | 34 | 5.3% | 5 | Hotel Facilities | HFC | 5 | 5.7% |
| 6 | Recreation and entertainment activities | HA | 34 | 5.3% | 5 | Wifi | HA | 5 | 5.7% |
| 7 | Hotel cleanliness | HFC | 32 | 5.0% | 6 | Poor management and policy | O | 4 | 4.6% |
| 8 | Staff helpfulness | S | 30 | 4.7% | 6 | Value for money | O | 4 | 4.6% |
| 9 | Room assignment | R | 29 | 4.6% | 6 | Room dirtiness | R | 4 | 4.6% |
| | | | | | 6 | Room assignment | R | 4 | 4.6% |
| Total | 29 | R:2, S:2, HFC:2, HA:2, O:2 | 386 | 54.6% | Total | 20 | R:3, S:2, HFC:1, HA:1, O:4 | 67 | 81.4% |

Note: Five hotel components (R: room, S: staff, HFC: hotel facilities and cleanliness, HA: hotel amenities, O: others).

In traditional accommodations, safety emerged as the foremost satisfier (9.3%), suggesting its critical role for solo female travellers. This was closely followed by location (8.9%) and staff helpfulness (8.9%), reflecting the importance of a convenient locale and a responsive support system for these guests. Other significant satisfiers include staff friendliness (7.4%), room cleanliness (7.4%), and breakfast (6.7%), emphasizing the necessity of warm interactions, clean rooms, and inclusive, often complimentary breakfast offerings for a satisfying hotel experience. Further, room amenities and bed comfort, contributing 6.3% and 5.9% respectively, underline the value of a well-furnished and comfortable accommodation space. Incorporating value for money (5.2%) and superior hotel facilities (4.8%), the top 10 satisfiers together account for 70.7% of all mentions, demonstrating that a well-rounded hotel experience is key to guest satisfaction.

Furthermore, we identified room dirtiness as the most prominent dissatisfier (16.9%). This factor was significantly ahead of insecurity (11.7%) and poor quality of room amenities (10.4%), suggesting a substantial emphasis on sanitation among solo female travellers. Other prominent dissatisfiers include staff unprofessionalism and harassment (8.4%), poor room condition (7.1%), and negative neighbourhood experiences (7.1%).

### 4.4. Satisfiers and Dissatisfiers in Alternative Lodgings

As outlined in Table 4, the top satisfiers in alternative accommodations in Canada, as per TripAdvisor reviews, highlight diverse areas of the guest experience. The most critical satisfier was found to be "location" (8.5%), trailed by "room amenities" (7.7%), and "staff friendliness" (7.4%). Other important satisfiers include "hotel facilities" (6.1%), "safety" (6.0%), and "recreation and entertainment activities" (5.3%). Interestingly, in alternative lodgings, "room amenities" (13.8%) emerged as the most significant dissatisfier, followed by issues with the "neighbourhood" (11.5%) and "service unavailability" (10.3%). Other notable dissatisfiers include "staff unprofessionalism and harassment" (8.0%) and negative experiences with "roommates and fellow guests" (8.0%). These findings suggest that the interplay between the physical and social environments profoundly impacts the experiences of solo female travellers in alternative lodgings.

### 4.5. Comparing Satisfiers and Dissatisfiers in Traditional and Alternative Hotels

Comparing satisfiers and dissatisfiers between traditional and alternative lodgings highlights distinct preferences of solo female travellers (see Figure 2). In traditional accommodations, key concerns are room cleanliness/dirtiness (10.8%), safety/insecurity (10.1%), room amenities (7.8%), and staff interactions, both helpfulness (6.6%) and friendliness (6.1%). The emphasis on cleanliness and staff indicates a preference for the predictable qualities of branded or affiliated hotel chains—a standard expectation for comfort in traditional settings. Additionally, the pronounced need for safety aligns with established findings in solo female travel research.

In contrast, alternative lodgings reveal slightly different priorities. While room amenities (8.4%), staff friendliness (6.9%), and safety (5.3%) remain important, there is less concern for cleanliness and staff assistance. Instead, location (7.5%) and hotel facilities (6.1%) take precedence. This suggests that solo female travellers in alternative accommodations are more flexible, valuing specific amenities and facilities over service and cleanliness, which indicates a readiness to adapt to a variety of lodging experiences.

Moving to the combined satisfiers in traditional and alternative settings, location (8.6%), staff friendliness (7.4%), room amenities (7.3%), safety (7.0%), and staff helpfulness (6.0%) emerge as the primary factors enhancing guest experiences. Conversely, room dirtiness (12.4%) ranks as the foremost dissatisfier, followed by inadequate room amenities (11.6%), unsatisfactory neighbourhood conditions (8.7%), staff unprofessionalism and harassment (8.3%), and a sense of insecurity (7.5%). These elements are critical in shaping the dissatisfaction of solo female guests.

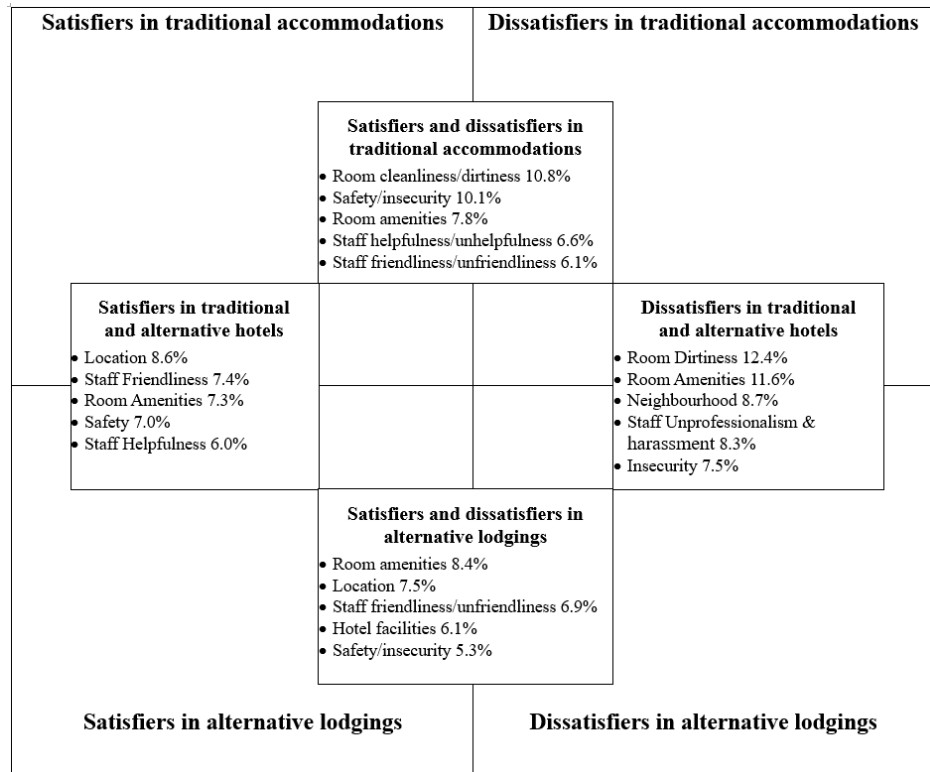

**Figure 2.** Comparison of satisfiers and dissatisfiers between traditional accommodations and alternative lodgings in Canada based on TripAdvisor reviews. **Note:** The figure displays only the top five satisfiers and dissatisfiers in each category.

*4.6. Average Number of Satisfiers and Dissatisfiers in Each Review*

We calculated the average count of satisfiers and dissatisfiers within satisfaction- and dissatisfaction-indicating reviews for both lodging types (see Figure 3). In traditional accommodations, the averages were 5.7 satisfiers and 3.8 dissatisfiers. For alternative lodgings, the averages were 6.6 satisfiers and 4.4 dissatisfiers per review. Our analysis reveals a higher average frequency of satisfiers and dissatisfiers in alternative lodgings compared to traditional accommodations. This could potentially be attributed to the diverse range of accommodation options that fall under the banner of 'alternative lodgings'. Such variety may lead to a wider spectrum of guest experiences and, consequently, an increased number of determinants for satisfaction or dissatisfaction. Furthermore, the less-standardized nature of these accommodations, in contrast to their traditional counterparts, might mean that guests have less consensus on what constitutes a standard experience, leading to a broader range of satisfiers and dissatisfiers. In conclusion, it appears that the task of satisfying guests is more challenging for alternative lodgings. However, these accommodations also seem to provide a richer array of satisfactory experiences, as indicated by the higher number of satisfiers identified in their reviews.

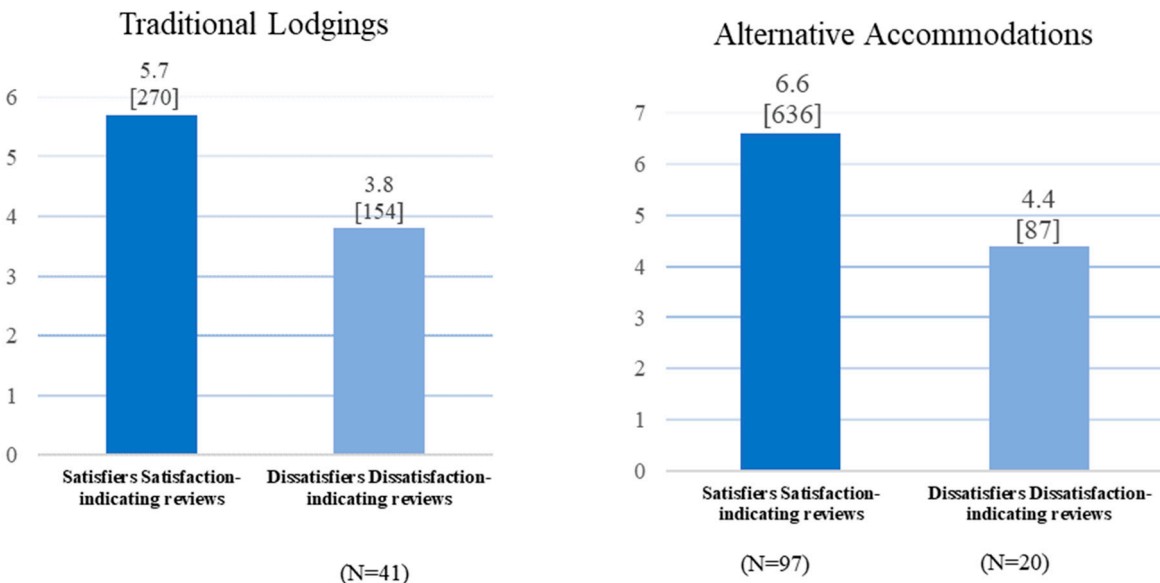

**Figure 3.** Average frequency of satisfiers and dissatisfiers per review. Note: Numbers enclosed within square brackets [] represent the total frequency of satisfiers or dissatisfiers.

## 5. Discussion

This study explored the factors that influence solo female travellers' satisfaction and dissatisfaction during hotel stays in a Canadian context, using online reviews of Canadian hotels from TripAdvisor. Employing Herzberg's two-factor theory, we identified a total of 29 satisfiers and 24 dissatisfiers in traditional and alternative lodgings. These factors were grouped into five main components: room, staff, hotel facilities and cleanliness, hotel amenities, and others. Furthermore, online reviews suggest that guests of alternative lodgings generally report higher satisfaction levels than those in traditional accommodations, though they also encounter a broader range of factors influencing their satisfaction and dissatisfaction.

### 5.1. Theoretical Contributions

This study enriches our understanding of solo female travellers' accommodation experiences, contributing significantly to the literature on solo female travellers [2,16], gender differences in hotel experiences [49,52], and customer satisfaction [25,40], with a special emphasis on the application of the two-factor theory [33]. Firstly, it enhances our knowledge of solo female travellers by analyzing online reviews of both alternative and traditional lodgings. This reveals the necessity for a more nuanced hotel segmentation approach beyond the conventional star rating or service level, as indicated by the significant proportion of reviews related to alternative lodgings. Valaja [80] discussed risk-reduction strategies employed by solo female travellers, such as choosing Airbnb accommodations for host support and selecting hostels to meet fellow travellers for companionship during international trips. Our findings align with this, as evidenced by the high percentage of reviews from alternative accommodations, reflecting these strategic choices. Furthermore, the satisfiers and dissatisfiers identified in this study align with the existing research on gender differences in hotel experiences [19,49]. This underscores the unique preferences and expectations of solo female travellers regarding hotel attributes and services, highlighting the importance of considering gender and travel companionship as critical factors in hospitality management.

We delineated five key components influencing guest experiences—room, staff, hotel facilities and cleanliness, hotel amenities, and others—based on the satisfiers and dissatisfiers identified through Herzberg's two-factor theory framework. These components significantly influence either satisfaction, dissatisfaction, or both. Understanding this dis-

tinction is vital for comprehending guest expectations, which differ between traditional and alternative lodgings. For instance, cleanliness and staff helpfulness are prioritized in traditional settings, while location and hotel facilities are more valued in alternative accommodations. However, safety, room amenities, and friendly staff are universally important across accommodation types. Thus, our study contributes to the theoretical discourse on the determinants of satisfaction and dissatisfaction in the hospitality industry [25,26,31], emphasizing the nuanced roles of various satisfiers and dissatisfiers.

*5.2. Practical Implications*

This study offers valuable insights for enhancing the accommodation experience of solo female travellers, a crucial step in ensuring equitable travel conditions and supporting greater gender equality in tourism [2]. The use of online reviews proves advantageous for the accommodation sector and other tourism industry operators [20,50,57], as it enables them to monitor and address the specific needs and dissatisfactions of solo female travellers promptly.

We identified key differences between alternative and traditional lodging experiences. Our findings suggest that solo female travellers in alternative accommodations seek unique experiences, differing from those offered by traditional hotels, and display higher satisfaction levels. This points to the need for distinct management strategies, balancing the opportunities and challenges presented by the higher number of satisfiers and dissatisfiers in each review. This aligns with the work of Radojevic et al. [40], who noted the demographic group effect on satisfaction levels, with solo travellers often reporting higher satisfaction compared to other groups.

The study's results, delineating specific satisfiers and dissatisfiers, serve as practical guidelines for managers to enhance service delivery. Common factors across both types include safety, room amenities, cleanliness, and friendly staff. For instance, addressing safety concerns, such as floor preferences for solo female travellers, is paramount. While technology and software can streamline room assignments, especially for group check-ins, they should be judiciously employed for solo female travellers. As Rahimi et al. [20] emphasize, understanding guests' preferences is vital due to differing responses to hotel services by gender. Hotel managers can leverage service encounters, like discussions on room allocation, to facilitate value co-creation and enhance guest experiences [81]. Therefore, this study not only contributes to the academic discourse but also provides actionable strategies for industry practitioners to optimize the lodging experience for solo female travellers.

*5.3. Limitations and Future Studies*

The limitations of this study encompass several dimensions. Firstly, the study's gender segmentation was limited to solo female travellers. Including the experiences of solo male travellers in future research [5] could substantially enhance our understanding of solo travel dynamics [16]. The categorization of accommodations was not exhaustive; traditional lodgings were not differentiated between full service and limited service [31], and alternative lodgings include, but are not limited to, Airbnbs, hostels, camping sites, glamping units and Inns and B&B's, highlighting the complexity of categorizing non-traditional accommodations. Future studies can benefit from a more detailed segmentation of accommodation types, as well as considering the interplay between sociodemographic factors such as age and income, and the identified satisfiers and dissatisfiers, including staff friendliness/unfriendliness. It is also crucial to note that solo travellers' budgetary concerns and expectations can significantly impact their hotel selections, determining the trade-offs they are willing to make. While our study did not specifically consider hotel attributes such as the city, service, location, and prices, these are all important factors for future studies to explore.

Although the existing literature largely links the motivations of solo female travellers to non-sexual factors [12,43]), recent research has started to examine the role of sexual

factors in solo female travel [82,83]. Notably, Frohlick [84] highlights a trend where western women engage with local men in non-western and developing countries, bringing attention to the phenomenon of female sex tourists. Future studies could explore how these sexual factors [85] influence their hotel selections and experiences. Geographically, this study's scope was limited and did not encompass a broader global context; given existing research on solo female travellers in countries like China [11] and Australia [14], there are significant opportunities for future exploration of their accommodation experiences in these regions.

Methodologically, the reliance solely on online reviews, particularly those containing keywords such as "solo female" or "single female", presents constraints. As solo travel can be motivated by "circumstances" or "choice" [16], some tourists might not employ these specific terms in their reviews, potentially leading to a selection bias. Future studies should employ diverse data-collection methods, including in-depth interviews, focus groups, or a mixed-methods approach to allow for a broader exploration of factors influencing solo female travellers' hotel satisfaction or dissatisfaction. Additionally, expanding the analysis to include a wider range of review platforms, beyond TripAdvisor, such as Expedia or Booking.com, could offer a more comprehensive understanding of guest experiences. Last, but not least, we did not capture the specific years of the 364 online reviews analyzed, which limits our ability to track changes in solo female travellers' preferences (such as wifi) and the evolution of hotel services over time. This gap highlights the potential for preferences and expectations, as well as the services and amenities offered by accommodations, to shift in response to industry trends and guest needs. As such, we recommend future research undertakes longitudinal analyses or comparative studies across different time periods with a larger dataset.

**Author Contributions:** Conceptualization, F.Z. and S.H.; methodology, F.Z. and S.H., software and formal analysis, F.Z.; validation, S.H. and M.M.; writing—original draft preparation, F.Z. and S.H., writing—review and editing, S.H. and M.M.; supervision, S.H. and M.M. All authors have read and agreed to the published version of the manuscript.

**Funding:** This research received no external funding.

**Institutional Review Board Statement:** Not applicable, as this study involved the analysis of publicly available online data.

**Informed Consent Statement:** Not applicable.

**Data Availability Statement:** The original contributions presented in the study are included in the article, further inquiries can be directed to the corresponding authors.

**Conflicts of Interest:** The authors declare no conflicts of interest.

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
