# Peer review of "Understanding Solo Female Travellers in Canada: A Two-Factor Analysis of Hotel Satisfaction and Dissatisfaction Using TripAdvisor Reviews"

_tourismhosp, doi:10.3390/tourhosp5010012_

Round 1
Reviewer 1 Report
Comments and Suggestions for Authors
This is a very interesting paper, on an emergent trend related to the changes and structural transformations currently under way western societies.
(1) The introduction section and the literature review are quite complete and instructive, full of examples and key conclusions from the literature. However, while it is acceptable to apply the Herzberg's two-factor theory to the analysis of the drivers and influencers of satisfaction, it would be advisable to add a few lines on the advantages of this theoretical approach vis a vis the alternative theoretical approaches mentioned in this section.
(2) It would be advisable to add figures (if available) to get an overview of the quantitative importance of this market segment. Complains about hotel´s lack of consideration, i.e., focus on couples and families, may lie in the lack of critical mass in terms of the number of solo travellers that may justify investments in this segment.
(3) About the methodological part. It is not clear how excluding “Reviews with a rating of 3” affect “the integrity of [the] data analysis”. As mentioned in the paper, even comments with a category 5 contain some negative comments, which forced the author(s) to “meticulously” analyse by hand such type of commentaries. In practice, it would be very relevant to get an insight on the thinking of those solo travellers with “mixed” feelings. I would suggest, that few lines on the broader picture resulting from such commentaries would add value to the paper.
(4) The results section in very interesting and illustrative. No major issues in this regard are detected, and the graphs and schemes are quite useful.
(5) One of the main limitations of this study lies in the lack of socio-demographic data, namely age and income. The sensitivity to Room cleanliness/dirtiness will be in all evidence different according to the initial expectations, in turn influenced by budgetary constraints and other reasons. Young solo travellers may be ready to trade-off some attributes for more opportunities to enrich themselves from a cultural point of view or to get more chances to contact other people. Therefore, the results must be read as generic conclusions, applying to an undifferentiated pool of female visitors. While such data in not directly available to the research team, information on the type of accommodation under analysis can include data on daily room prices, cost of living issues, distance to down-town area or other key points, etc, that can be used to infer about the impact of issues such as prices and budgetary constraints on the individual´s assessment of attributes such as cleanliness/dirtiness.
(6) As mentioned in the paper, “future studies can benefit from a more detailed segmentation of accommodation types, provided there is sufficient data for robust analysis”. For the moment, it could help the reader to add a few notes, based on other studies, on the impact of variables such as Age on issues such as Staff friendliness/unfriendliness.
Author Response
Thank you for your thorough review and valuable feedback. Please see the attached document for our detailed responses to your comments.

Reviewer 2 Report
Comments and Suggestions for Authors
This paper investigates the factors that influence solo female travellers’ satisfaction from hotel stay in Canada. The idea of this paper is interesting and contributes to a topic of increasing interest in recent years (solo travelling). However, the authors are invited to revise the manuscript to address specific concerns before a final decision is reached. The following are needed in the paper if it is to be resubmitted:
· Lines 54-57: The authors describe the gap in the literature they are trying to address. The authors need to write purpose of the paper in a way that will justify the unique contribution of this paper. The originality statement of the article should be clearly articulated by supporting more theoretical clarification/evidence as well as rationale why should the reader be interested in.
· Line 37: Please refer to the SoFe Traveller Network with its full name (The Solo Female Traveler Network).
· Lines 128-149: The authors analyze the motivations of solo female travelers. What about female sex tourism? There is a large body of literature focusing on this topic that has been ignored by the authors.
· In the research methodology, the researchers do not mention which years the reviews they analyzed are from. What year do the reviews start? This is an important factor as the preferences and behavior of tourists change over time. For example, the provision of free Wi-Fi by hotels that is now considered extremely important by customers a few years ago did not even exist as a service provided by hotels.
· Lines 308-318 and 350-441: This information describes research steps and should be transferred to the methodology section (3).
· Lines 551-554 and 577-579: There is repetition of the same information.
· Lines 279-281: The authors argue that B&Bs, university residence, camping sites, hostels, and other accommodations were classified as alternative lodgings while on line 623 they state that alternative lodgings are limited to Airbnb, hostels, Inns & B&Bs. What does alternative lodgings include? Are there Airbnb reviews on TripAdvisor or are they only on the Airbnb platform?
· The manuscript would benefit from a professional editing and proofreading. There are some grammatical and syntactical errors in the text, e.g. lines 29-31 (“recent times” and “recent years” in the same sentence), line 552 (in in).
· Authors are encouraged to follow journal’s instructions for references in text (numbered) and reference list.
·
Comments on the Quality of English LanguageMinor editing of English language required
Author Response

(The authors gave the same response as above.)

Reviewer 3 Report
Comments and Suggestions for Authors The paper presented is interesting, with solo female travellers being a quantitatively non-negligible trend and, therefore, its analysis is important. In recent years, many solo female travellers as a personal experience that allows them to learn more about the places visited and meet new people during the trip, but this type of trip has its specific peculiarities and characteristics. This paper is an advance in the knowledge of the aspects that generate satisfaction and dissatisfaction for this type of travelers. Being a relatively recent topic, there is still little specific literature and it is necessary to generate a significant volume of studies before generalizable conclusions can be generated. This paper makes an interesting contribution in this sense. The presentation of the problem and the review of the literature is very complete and well presented, with the references being appropriate and quite current. On the other hand, the results are quite detailed, exposing the numerical data in tables that facilitate the consultation. The methodology used has limitations that are discussed in the limitations section, but it provides an interesting approach to the topic of study. The only drawback I would have is that the data analysis could be a little more detailed to facilitate its replicability in future studies. In particular, the analysis carried out once the final sample of the comments that will be analyzed is obtained and how it arrives at the concepts and data that appear in the results tables should be better detailed. The conclusions are adjusted to the analyzes carried out and the results obtained, exposing the theoretical and practical implications of the results, in addition to indicating the limitations derived from the methodology used.
Author Response

(The authors gave the same response as above.)

Round 2
Reviewer 1 Report
Comments and Suggestions for Authors
Dear author(s)
Many thanks for taking into consideration the suggestions. I think that the paper´s quality improved. Congratulations. All the best.
Reviewer 2 Report
Comments and Suggestions for Authors
The reviewer’s comments have been addressed. The text has been improved in quality. My recommendation is that the manuscript be accepted for publication.
Comments on the Quality of English LanguageMinor editing of English language required
Reviewer 3 Report
Comments and Suggestions for Authors
The previously indicated indications for improvement have been addressed in a reasonable manner.